

# Thermal summation model and instar determination of all developmental stages of necrophagous beetle, *Sciodrepoides watsoni* (Spence) (Coleoptera: Leiodidae: Cholevinae)

Pavel Jakubec

Department of Ecology, Faculty of Environmental Sciences, Czech University of Life Sciences Prague, Praha, Suchdol, Czech Republic

## ABSTRACT

Necrophagous beetles are underrepresented in forensic entomology studies despite their undeniable utility for the field. In the present article, information is presented regarding the developmental biology and instar determination of *Sciodrepoides watsoni* (Spence, 1813), a very common species occurring across the Holarctic region. Wild collected beetles were kept in climate chambers at constant temperature (12, 15, 18, 21 and 28 °C) and their development was regularly documented. Parameters of thermal summation models and standard errors were calculated for each developmental stage. These models may be used for an estimation of post-mortem interval in legal investigations after further validation on local populations of *S. watsoni*. An additional methodology is introduced for future studies of size-based characteristics, addressing instar identification bias. The methodology provided estimations (mean, standard error and standard deviation) of *S. watsoni* larval head capsule width for preliminary larval instar determination. The methodology may be used with other morphological features to improve instar determination accuracy.

## INTRODUCTION

Forensic entomology is a rapidly developing new field of science (*Midgley, Richards & Villet, 2010*). New methods and models for estimation of minimum post-mortem interval (PMImin) are developing at a very rapid pace (e.g., pre-appearance interval, gene expression during larval development, quantile mixed effects models, generalized additive modeling or generalized additive mixed modeling) (*Matuszewski, 2011*; *Tarone & Foran, 2011*; *Baqué et al., 2015a*; *Baqué et al., 2015b*), but even the well-established models lack actual data for their further use and application. A good example is the commonly used thermal summation model (*Richards & Villet, 2008*). This model, which is based on the assumption that development of immature stages is linear, has been known for several decades (*Higley, Pedigo & Ostlie, 1986*), but it is still not established for the majority of forensically important species of invertebrates, which would be a great contribution to legal investigations.

Corresponding author
Pavel Jakubec, jakubecp@fzp.czu.cz

Currently these models are known for a number of fly species (Diptera) (*Nabity, Higley & Heng-Moss, 2006*; *Villet, MacKenzie & Muller, 2006*; *Richards, Crous & Villet, 2009*; *Voss, Spafford & Dadour, 2010a*; *Voss, Spafford & Dadour, 2010b*; *Voss et al., 2014*; *Tarone et al., 2011*; *Nassu, Thyssen & Linhares, 2014*; *Zuha & Omar, 2014*), but only for a few necrobiont beetles.

However, using beetles for PMImin estimation has several benefits compared to flies. Beetles tend to have a longer development, and therefore can be found on and around the carrion for a longer period of time (*Villet, 2011*). They also do not form a maggot ball like flies, and they can be reared individually so they are easier to handle in laboratory conditions (*Midgley, Richards & Villet, 2010*). However, probably the best advantage is the possibility of cross validating PMImin estimates between species and groups, such as flies and mites. Cross validating is important mainly in cases when one of these groups or species is affected by external factors (restricted access to body, temperature too high or low, etc.) providing a biased estimate (H Šuláková, pers. comm., 2014).

Statistically robust thermal summation models are available only for three species of necrophagous beetles, all belonging to the family Silphidae. These models are for *Thanatophilus micans* (Fabricius, 1794) (*Ridgeway et al., 2014*), *T. mutilatus* (Castelnau, 1840) (*Ridgeway et al., 2014*) and *Oxelytrum discicolle* (Brullé, 1840) (*Velásquez & Viloria, 2009*). *T. micans* occurs mainly in Africa and extends to Yemen on the Arabian Peninsula (*Schawaller, 1981*; *Růžička & Schneider, 2004*), *T. mutilatus* has a geographical distribution restricted to the South Africa region (*Schawaller, 1981*; *Schawaller, 1987*) and *O. discicolle* inhabits Central and South America (*Peck & Anderson, 1985*). Therefore, North America, Europe and most of Asia lack a single beetle species with a thermal summation model.

Models alone are not sufficient for species to be useful in legal investigations. There are other criteria to be fulfilled. Any forensic entomologist has to be able to identify those species in every stage of development and discriminate between larval instars. Without reliable instar determination, it is not possible to expect reliable PMImin estimates. However, this is sometimes complicated, because beetle larvae often lack morphological characteristics specific to particular instars, which would allow such identification (*Velásquez & Viloria, 2009*). In such cases the body size frequency distributions are commonly used (see *Logan et al., 1998*).

Not all size-based characteristics are equally appropriate for instar determination. *Dyar (1890)* observed that head width is a very advantageous characteristic for larval instar determination. He supported his claim by showing that head width stays the same across intermoult intervals and also that for some lepidopteran larvae it follows a geometrical progression. These properties enable estimation of number of instars even from incomplete developmental information, when some instars were unobserved and are widely used for instar determination (e.g., *Delbac, Lecharpentier & Thiery, 2010*; *Wu, Wang & Wu, 2012a*; *Wu, Wang & Wu, 2012b*; *Gomez et al., 2015*).

Head width or mean head width can be found for many species in their published descriptions together with other morphological and size based characteristics. Unfortunately these measurements are often based on a small number of animals, because they are not intended as statistically robust models, thus the true mean value could be very

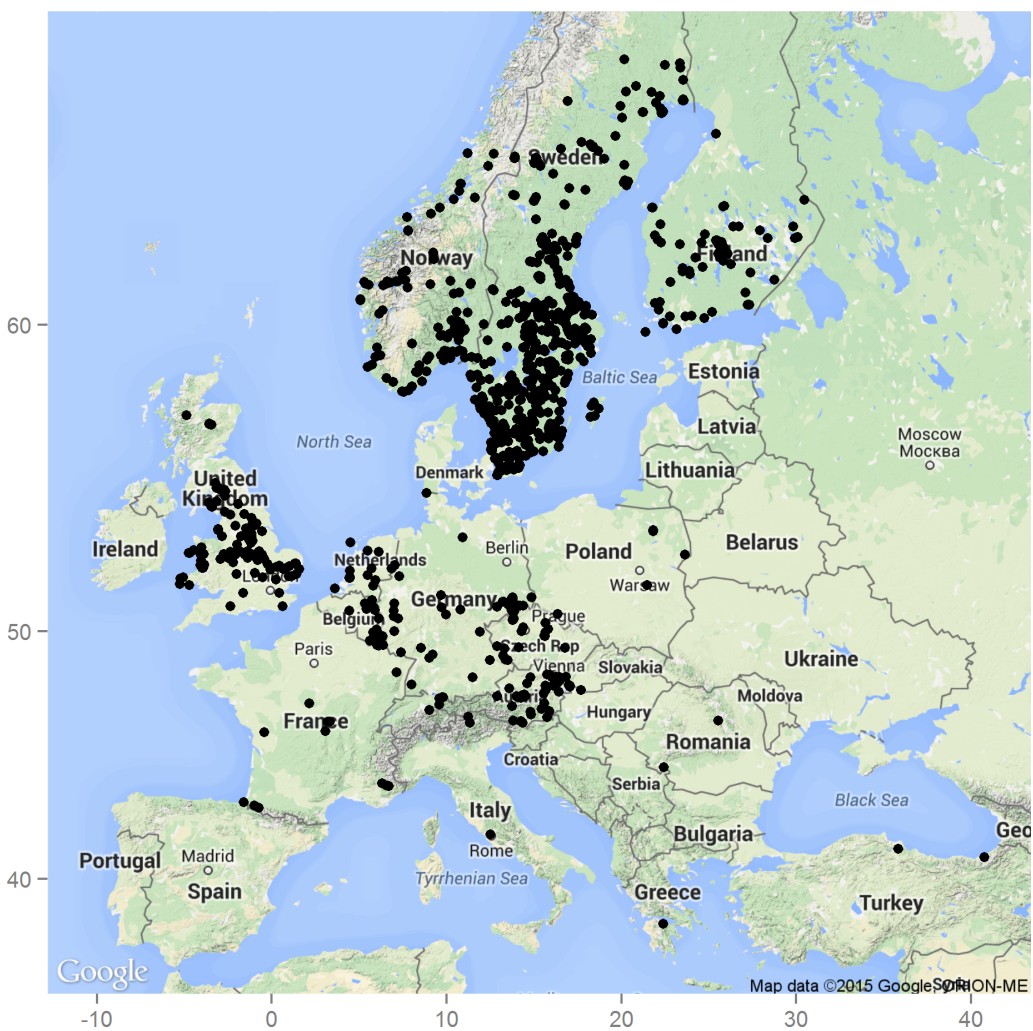

**Figure 1** **Occurrence of *S. watsoni* in Europe based on observations and records from the GBIF database (*GBIF, 2015*).** Underlying map generated by package ggmap (*Kahle & Wickham, 2013*). Map data ©2015 Google, ORION-ME.

different from the reported one. This bias is a serious problem for instar determination in applied disciplines like forensic entomology, especially when we are dealing with animals from spatially or temporally distant populations (*Stillwell & Fox, 2009*). For several species of necrobiont beetles statistically robust models were developed, based on measuring not only head width, but also other size based characteristics (*Midgley & Villet, 2009b*; *Velásquez & Viloria, 2010*; *Fratczak & Matuszewski, 2014*). However, these models are available only for two European species, namely *Necrodes littoralis* (Linnaeus, 1758) (Silphidae) and *Creophilus maxillosus* (Linnaeus, 1758) (Staphylinidae) (*Fratczak & Matuszewski, 2014*) and even those models should be used with great care as I will discuss later in the text.

*Sciodrepoides watsoni* (Spence, 1813) is one of the most widespread and abundant species of necrophagous beetles in the Holarctic region (*Peck & Cook, 2002*; *Perreau, 2004*). Robust occurrence data are available especially for Europe (Fig. 1). This saprophagous

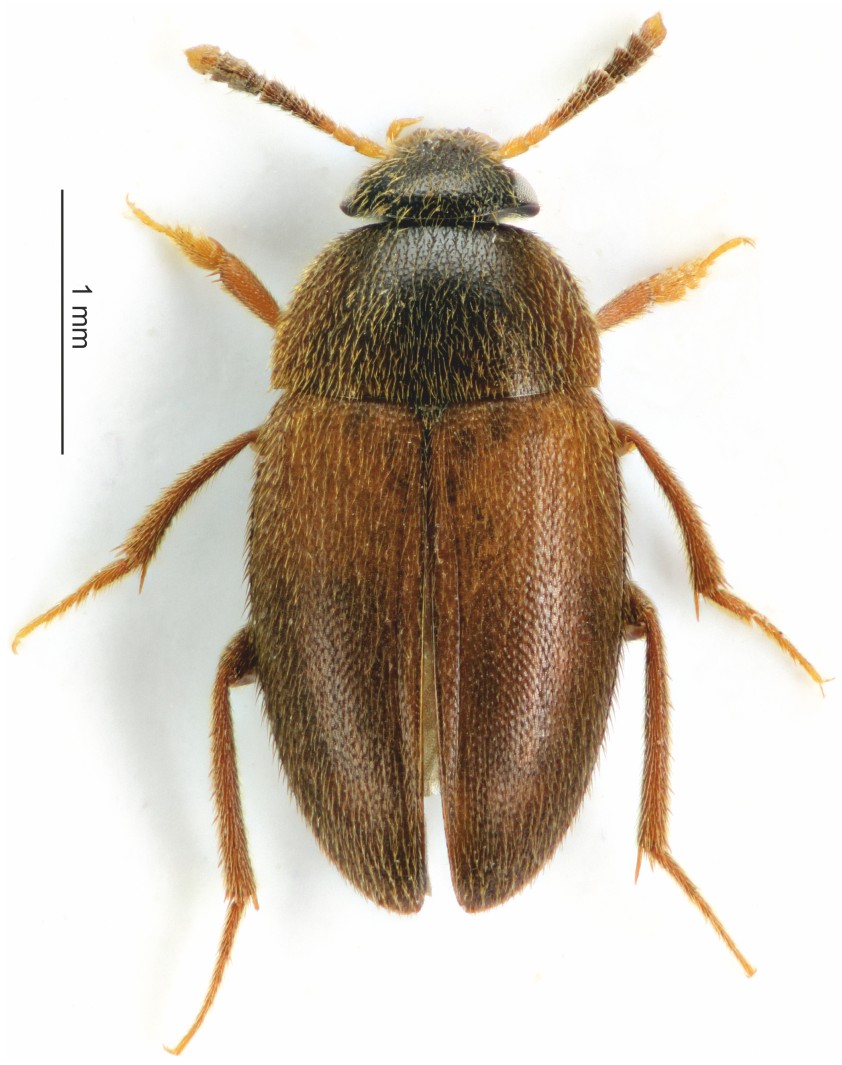

**Figure 2 Habitus of the *S. watsoni* male from dorsal view.**

beetle belongs to subfamily Cholevinae (Leiodidae) and is rather inconspicuous, because the whole body is brown and about 3 millimeters long (*Szymczakowski, 1961*; *Perreau, 2004*) (see Fig. 2). Adults can be fairly easily distinguished from the other European species of genus *Sciodrepoides* by the shape of the antennal segments (*Szymczakowski, 1961*). The main peak of activity is during the warmer parts of the year (late spring and summer) (*Růžička, 1994*). All stages can be found on decaying corpses of vertebrates in various types of habitats where they feed and develop (*Růžička, 1994*; *Peck & Cook, 2002*; *Topp, 2003*).

Egg, all larval instars and pupae of *S. watsoni* were described recently by *Kilian & Mądra (2015)*, and also a DNA barcode for validation is available (*Schilthuizen et al., 2011*). Therefore identification of this species in every stage of development is not an issue. Instar determination of *S. watsoni* larvae is also possible (*Kilian & Mądra, 2015*), but its natural variability was not covered in the case of size based characteristics.

This study attempts to improve the utility of *S. watsoni* for PMImin estimation by calculating the parameters of thermal summation models for each stage (egg, three larval instars and pupae) and developing an additional characteristic for instar determination, based on photographic documentation and measurement of larval head capsule width. The latter methodology may be developed to cover natural variability and can be easily observed, measured, and evaluated. Combined with a morphological feature unique to specific instars, these data provide accurate identification of larval instar and may be integrated into PMImin estimation models.

## MATERIAL AND METHODS

A laboratory colony was started with adults of *S. watsoni*, which were collected in spring of 2012 and/or 2013 from five localities in the Czech Republic (Prague–Suchdol (15 May–12 April 2012, 15 May–12 April 2013), Běstvina (7–11 April 2012, 6–10 April 2013), Domažlice (28 May–12 April 2013) and Klatovy (14–28 May 2013)).

Beetles were collected using 10 baited pitfall traps, placed at each locality. The traps were composed of 1,080 ml plastic buckets (opening of 103 mm and 117 mm deep). These buckets were embedded in substrate up to the rim to eliminate any obstructions which could deter beetles from entering. As protection against rain metal roofs (150 × 150 mm) were put over the traps. The roof was supported by four 100 mm nails, one in each corner, and placed approximately two centimeters above the surface. The bait, ripened cheese (Romadur®) and fish meat (*Scomber scombrus* Linnaeus, 1758), was placed directly inside the bucket on a shallow layer of moist soil. This created excellent conditions for the survival of the trapped beetles between servicing, which was usually done once a week.

After transport to the laboratory, beetles were identified and sexed under a binocular microscope (Olympus SZX7). Most of the beetles were randomly assigned to form breeding groups of at least four individuals (two males and two females). Specimens from the same locality were kept together regardless of capture date to eliminate cross-breeding of different populations. These groups were formed to produce new progeny, which were observed throughout their development (breeding experiment).

Groups were kept in Petri dishes with a layer of soil and small piece (approx. 5 × 5 mm) of fish meat (*Scomber scombrus*) as a food source. The content of the dish was lightly sprayed with tap water every day and food was provided *ad libitum* and changed to prevent fungal growth.

Dishes were randomly placed in one of six climate chambers (custom made by CIRIS s.r.o.). The chambers were set up at constant temperature (12, 15, 18, 21, 25 or 28 °C) and 16 h of light and 8 h of dark photoperiod regime, maintained by fluorescent light (Osram L 8W/640). A similar number of breeding groups from the same locality were placed in each chamber for beetles from Praha and Běstvina. However, because few adults were obtained from Domažlice and Klatovy, this was not possible, and they were kept together in one treatment (18 °C).

An observation study of their natural behavior was also conducted in a small plastic box (15 × 6 × 2 centimeters) with 12 adult individuals (seven females and five males) from the Prague population. In this colony, larvae were not separated from adults or each other,

but were allowed to interact freely and without intervention (measuring, photographing or other manipulations). The box itself was placed in an 18 °C treatment, and its inhabitants were attended in the same way as the specimens in the breeding experiment (regular water spray and meat replaced to prevent fungal growth).

In the breeding experiment, the method of handling eggs and first instar was changed slightly between the years to improve the accuracy of observations. During the first year of the experiment (2012), dishes were searched for eggs which were transferred individually to a separate dish. However, it was difficult to locate eggs of *S. watsoni* as they were very small and adults tended to hide them in the substrate. Therefore, the estimation of egg and L1 development for the first year was inconsistent and was not used in the models. In the second year (2013), the entire breeding group was transferred to a new Petri dish every day. The old dishes were marked and kept in the same climate chamber as the parents. Dishes were checked every day for emergence of the first instar larvae that were further separated into their own dishes so that individual development could be observed. The time when the eggs were laid was estimated as a half-time between the transfers of the breeding group.

Larvae from the second year (2013) breeding experiment were photographed every day, starting with their occurrence as the first instar larvae until pupation. In this way, morphological changes were continuously documented during their development. To do this, the Petri dish was removed from the climate chamber and was placed under the stereoscopic microscope to locate the larva, which tended to stay near the food source. The larva was transferred with a fine brush to a white sheet of paper and photographed (microscope Nikon SMZ-2T with attached camera Sony with resolution of 768 × 576 pixels). Once a usable picture was obtained, the larva was returned to Petri dish and back to the corresponding climate chamber. The whole process of finding the larva and taking a picture did not usually take more than 1 min. Key developmental stages of each larva, with the accurate date and time, could be distinguished based on photographs simply by keeping track of the change in the width of their head capsule, because the width expanded after each molt. An increase in the width over 0.1 mm was considered as a clear sign of the molt. This strategy was very useful for data collection, because exuvia were not needed for confirmation of molt to the next instar.

Because the dorsal side of larvae was photographed daily, many characteristics were monitored. However, the thorax and abdomen of the *S. watsoni* larvae are not strongly sclerotized (Fig. 3) and were thus omitted for instar determination, as well as the body length, which has too much variation. Measuring of some smaller parts, such as urogomphi or antennae, proved impractical, because it was very challenging to measure them accurately on a living and moving animal. Chilling or $CO_2$ immobilization could not be used, because it would stress the specimens even more and it could potentially affect the length of development.

The most stable and reliable feature for the instar determination of *S. watsoni* larvae was the head capsule. This part of the body was strongly sclerotized, therefore it was not affected by water or food content, but it changes in size after each molt so it is tightly linked with individual growth. Also, the head does not change its size in different fixation media, or even after desiccation, thus the instar can be identified even for very poorly handled and

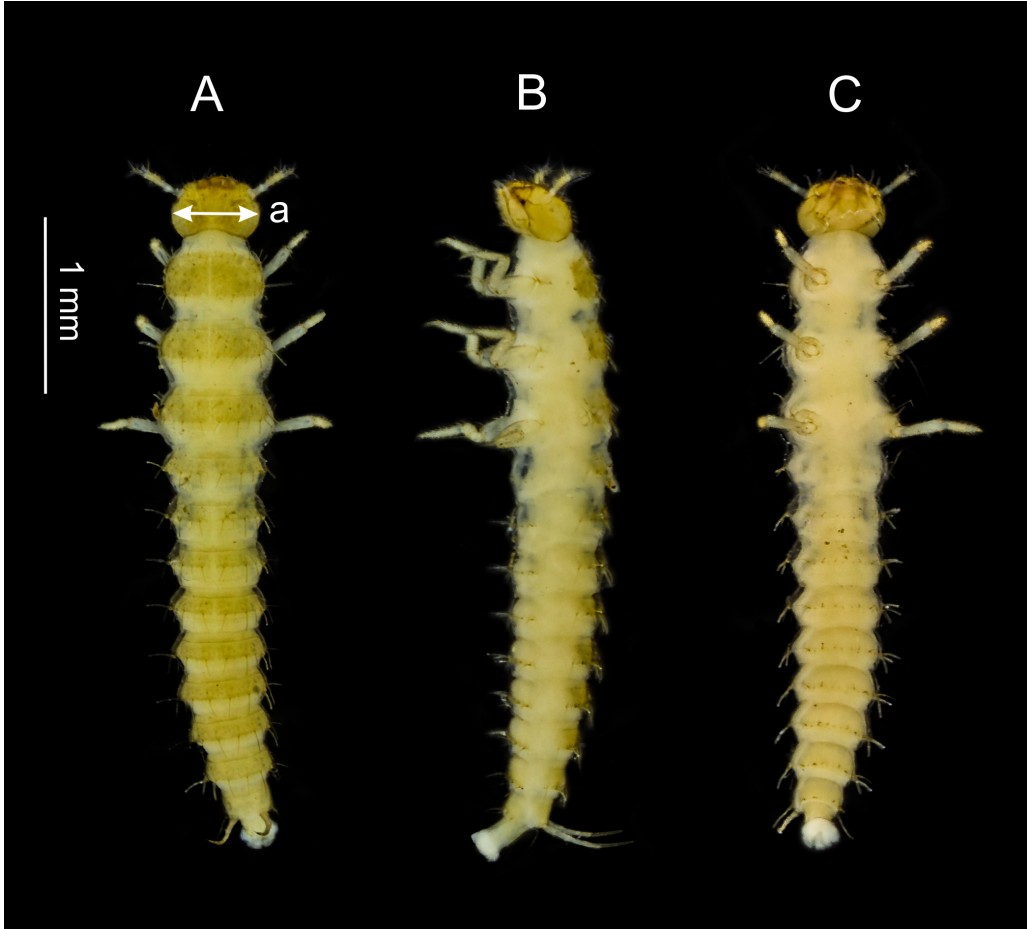

**Figure 3** **Dorsal (A), lateral (B) and ventral (C) side of the third larval instar of *S. watsoni*.** The point where the head width was measured (the widest point of the head capsule) is shown (a).

long dead specimens. Ultimately, the head capsule width was chosen over its length for a practical reason. The head width of living larvae did not change on the pictures captured from above, but the length varied considerably.

For estimating the mean and standard deviation of the head capsule width (measured in the widest point, see Fig. 3), I used all photographs where the head was clearly visible and was sharp enough to make a precise measurement. All measurements were with graphical program EidosMicro, calibrated by a precise ruler.

Parameters of thermal summation model (lower developmental threshold ($t$) and sum of effective temperatures ($k$)) were estimated for each developmental stage using the major axis regression method (($DT$) $= k + tD$), where $D$ is duration of development, and $T$ is environmental temperature (°C). This formula was developed previously (*Ikemoto & Takai, 2000*) and is commonly used for estimation of thermal summation parameters and their standard errors in forensic entomology (e.g., *Midgley & Villet, 2009a*; *Ridgeway et al., 2014*). The method is based on a standard linearized formula ($1/D = -(t/k) + (1/k)T$), but it weights out the data points in lower and upper part of the temperature range to obtain more reliable estimates of the parameters.
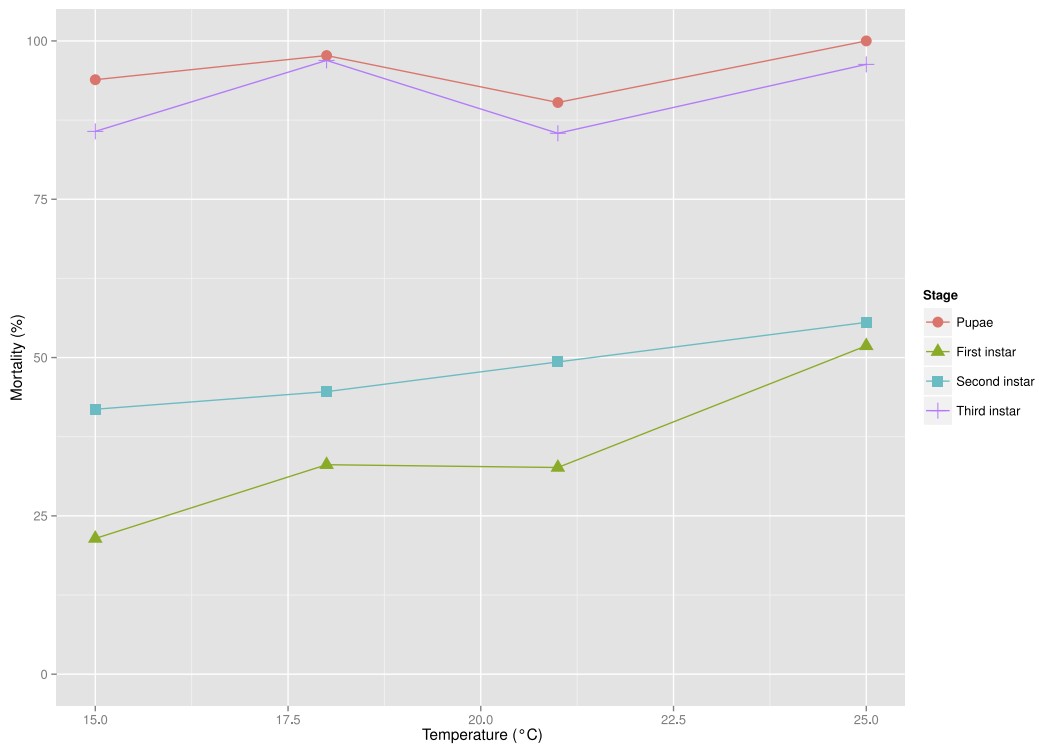

**Figure 4** Mortality rates between developmental stages of *S. watsoni* over a range of experimental temperatures, except for 12 and 28 °C, where beetles did not breed successfully.

Normality of all the data was confirmed by evaluation of the q-q plots and histograms. The significance level was set at 5%. Data management and all analysis were carried out using the R statistical program (*R Core Team, 2015*). Graphical outputs were handled by ggplot2 and ggmap R packages (*Wickham, 2009*; *Kahle & Wickham, 2013*).

## RESULTS

In total, 81 adult specimens of *S. watsoni* were collected (Prague—174, Běstvina—178, Klatovy—19, Domažlice—28), and they produced 399 first instar larvae for the breeding experiment. Because only twelve adults were obtained from Klatovy and six from Domažlice, it was impossible to split them between temperature treatments, and they all were reared at 18 °C.

In the breeding experiment, the duration of development of all *S. watsoni* stages was recorded, namely egg, three larval instars (L1, L2 and L3) and pupae. These observations were made on 399 specimens in total, starting with the first instar larvae.

Higher temperatures (25 and 28 °C) were probably limiting to breeding activity of the beetles in the experiment. Ultimately I did not obtain any larvae from the 28 °C treatment. Mortality in the other treatments was also quite high, especially for the third instar and pupae (see Fig. 4) and only 23 individuals developed until adulthood. The low temperature also prevented breeding, as I did not observe any larvae in the 12 °C treatment.

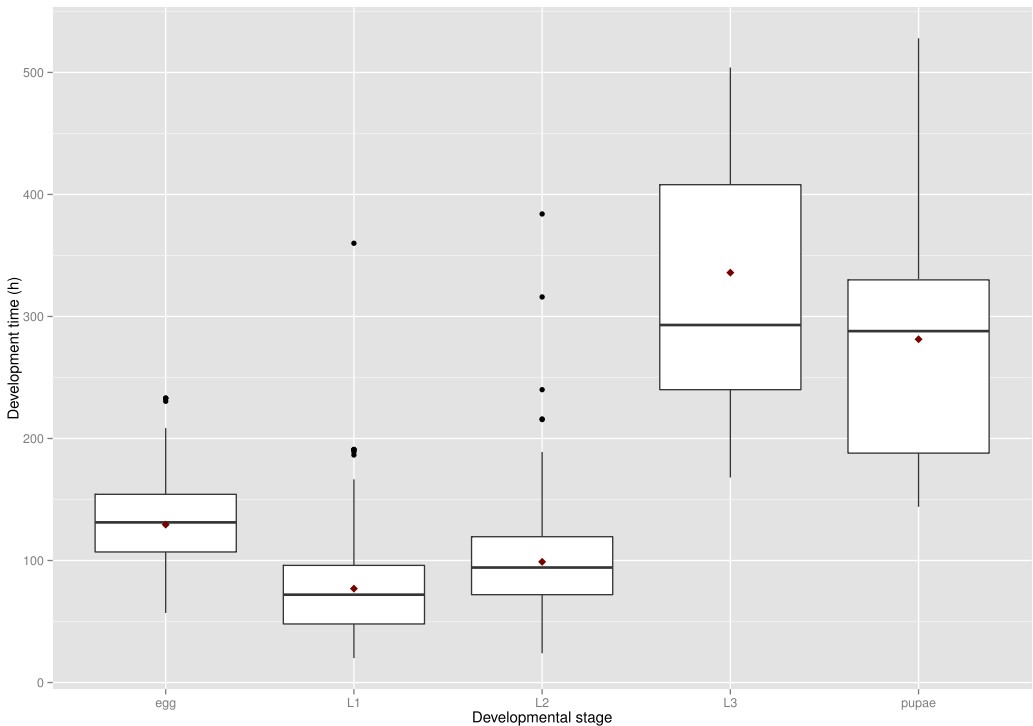

**Figure 5** **Observed range of development times of *S. watsoni* over four experimental treatments (15, 18, 21, 25 °C) for each developmental stage (2012 data were excluded for egg and L1).** The horizontal lines within the boxes indicate median values. The upper and lower boxes indicate the 75th and 25th percentiles, respectively. Whiskers indicate the values with the 1.5 interquartile ranges. Small, black dots are outliers. Small red dots are the mean values of development time.

**Table 1** **Summary of development constants for *S. watsoni* for five developmental stages.** Sum of effective temperatures ($k$) and lower developmental threshold ($t$) shown as means with standard errors (coefficient of determination ($R^2$) and degrees of freedom ($Df$) and $p$ values are provided).

| Stage | Temperature range | $R^2$ | $Df$ | $p$ value | $k$ | $t$ |
|-------|-------------------|-------|------|-----------|-----|-----|
| Egg   | 15–25 | 0.8134 | 220 | 2.20E–16 | 929.354 ± 49.111 | 11.400 ± 0.368 |
| L1    | 15–25 | 0.9375 | 171 | 2.20E–16 | 233.683 ± 27.031 | 15.437 ± 0.305 |
| L2    | 15–25 | 0.8768 | 206 | 2.20E–16 | 243.945 ± 45.301 | 15.689 ± 0.410 |
| L3    | 15–25 | 0.8199 | 27  | 1.49E–11 | 2602.996 ± 297.464 | 9.375 ± 0.846 |
| Pupae | 15–21 | 0.8563 | 10  | 1.61E–05 | 1207.431 ± 489.288 | 12.535 ± 1.624 |

The development times differed between stages (Fig. 5) and the mean development time decreased with increasing temperature (Fig. 6), except for L2 and L3 instars in the 25 °C treatment. The sum of effective temperatures ($k$) and lower developmental threshold ($t$) values were calculated for all developmental stages of *S. watsoni* with expected errors (Table 1 and Fig. 7).

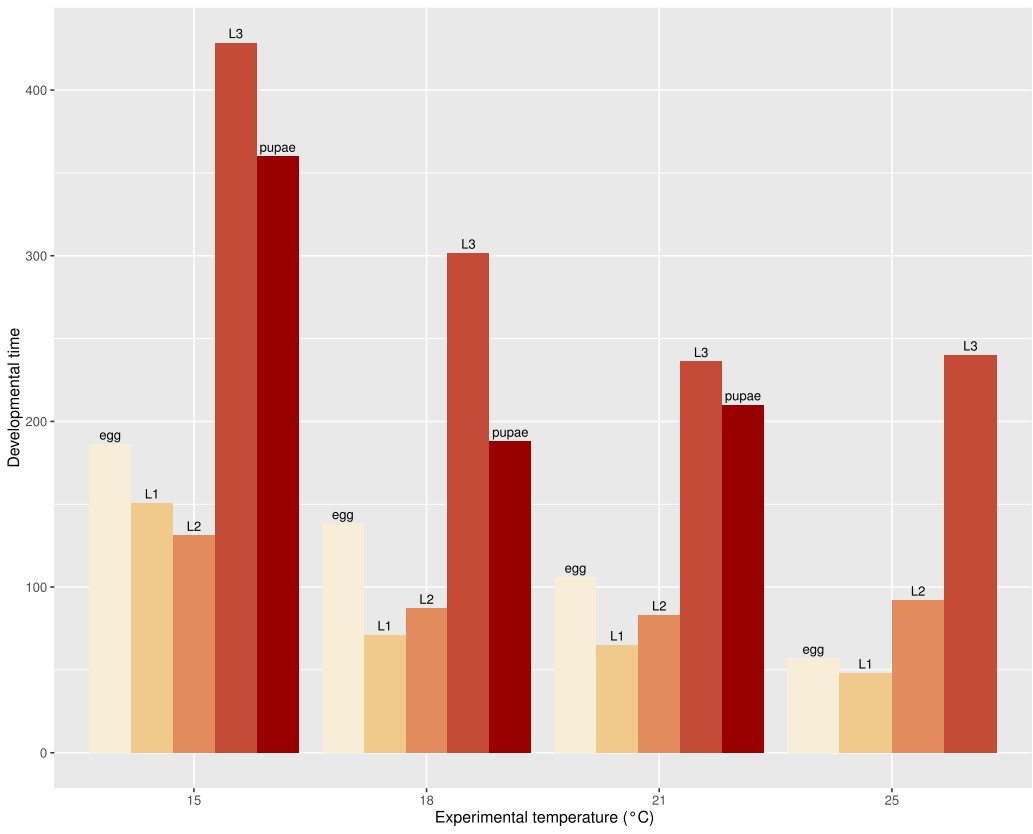

**Figure 6** Bar plot of mean development time (in hours) of all observed stages (2012 data were excluded for egg and L1) of *S. watsoni* over a range of experimental temperatures, except for 12 and 28 °C, where beetles did not breed successfully.

The mortality of the specimens in the observation study could not be measured, but the colony itself prospered very well and number of adults increased steadily, which is in contrast with what I observed in the breeding experiment. Females tended to hide their eggs in small holes or crevices in the substrate. Newly hatched larvae could be found mostly around the food source. The third instar larvae, after few days of feeding, dug underground and created small chambers where they pupated. No cannibalism or hostility of any kind between individuals was recorded.

For the instar determination measurements, I made 2,104 photographs, but only 1,731 were good enough to allow precise measurements of the head width. Those pictures covered all three larval instars (L1 = 591, L2 = 500 and L3 = 640 pictures). The bias in the number of pictures between different stages was caused by the difference in the duration of development of these instars (lower stages of development are shorter in duration), and it was also much more challenging to take a usable picture of the first or second instar larvae.

The mean width of the head capsule was a good additional characteristic for the instar determination (see Table 2 and Fig. 8). Standard deviations were well separated, and there was only a small overlap between 75th and 25th quintiles across all instars. I recorded some extreme values on the both sides of the spectrum, but those were very rare. If head capsule

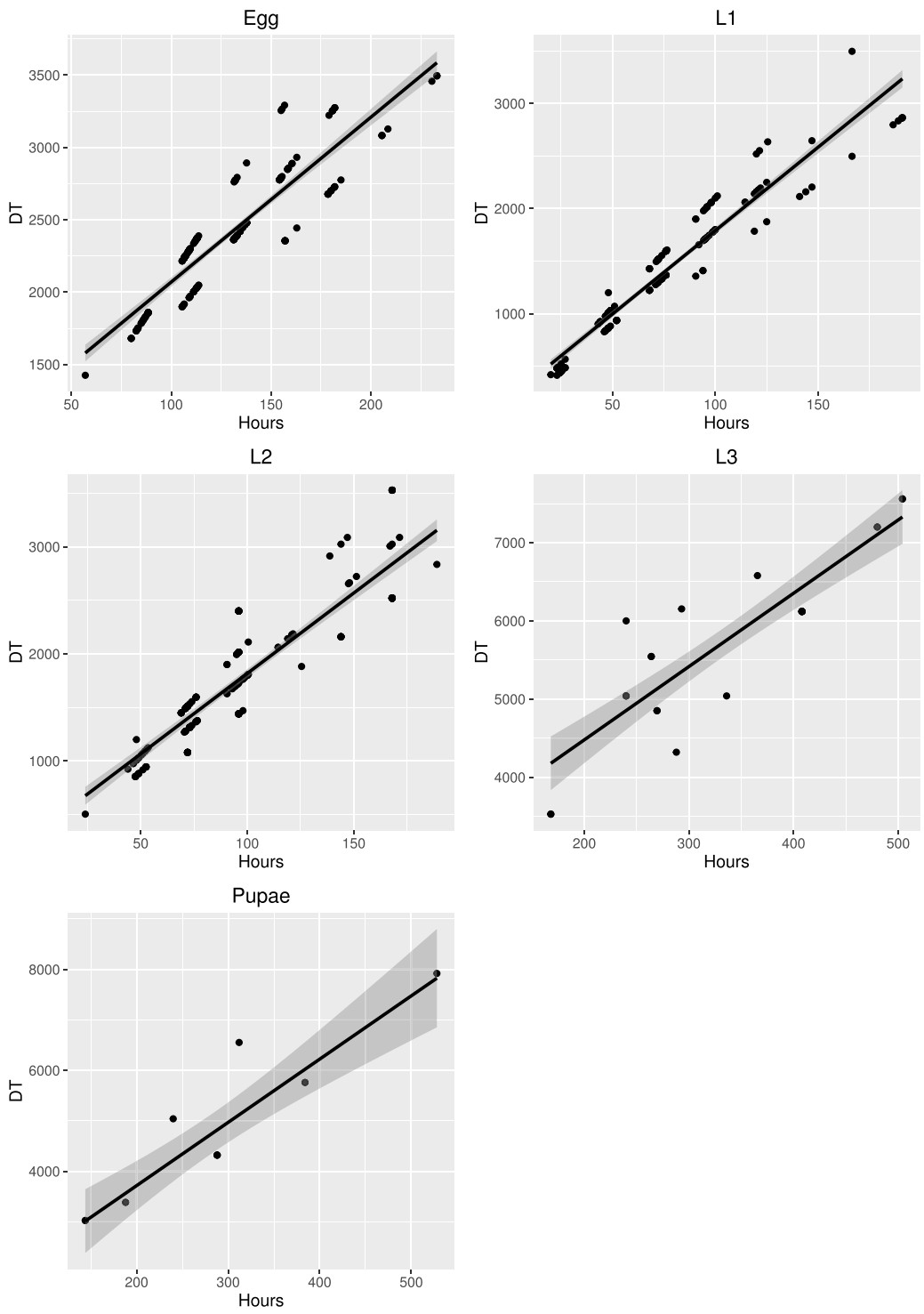

**Figure 7  Major axis regression for all stages of development in *S. watsoni*.** The black line shows median and gray area around is standard error. DT is the time in days to reach the stage multiplied by the constant rearing temperature. The 2012 data were excluded for egg and L1.

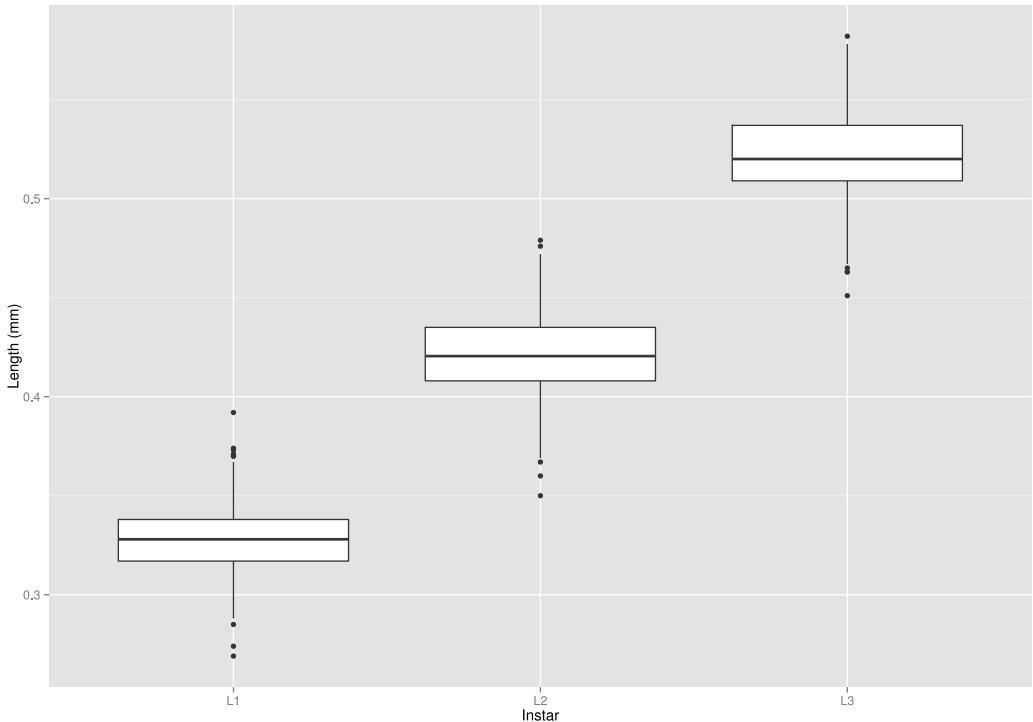

**Figure 8** **Box plot graph of lengths of all three instars (L1, L2 and L3) of the *S. watsoni* larvae.** The horizontal lines within the boxes indicate median values. The upper and lower boxes indicate the 75th and 25th percentiles, respectively. Whiskers indicate the values with the 1.5 interquartile ranges. Small black dots are outliers.

**Table 2** **The head widths (in millimeters) of all three larval instars of *S. watsoni*.**

| Instar | Max. | Min. | Mean | Stand. dev. |
|--------|-------|-------|-------|-------------|
| L1 | 0.392 | 0.270 | 0.329 | 0.017 |
| L2 | 0.479 | 0.350 | 0.421 | 0.021 |
| L3 | 0.582 | 0.451 | 0.522 | 0.021 |

measurement is used along with morphological characters like chaetotaxy and brown spot on the head, as described by *Kilian & Mądra (2015)*, the accuracy and precision of larval instar determination of *S. watsoni* may be improved.

## DISCUSSION

No larvae were obtained from the 28 and 12 °C treatment, probably because adults did not oviposit at this temperature or egg mortality was too high. The second claim is supported by the fact that no eggs were found. However, as mentioned in the methodology section, eggs of *S. watsoni* are tiny and may be overlooked, especially if there were only a few.

Mortality of the specimens in the breeding experiment was very high over all treatments especially in the third instar. High mortality in this stage is not uncommon, but in this case it was in sharp contrast with what was found in the observation study. The entire colony

in the observation study prospered and even increased in the number of adults over time. The only difference between these two studies was that individuals were not separated and photographed in the observation study.

Despite the increased mortality of some stages in the breeding experiment, the total length of development (from egg until adulthood) did not differ significantly from values in the observation study (ca. 28 days at 18 °C) and also those reported by *Kilian & Mądra (2015)* (ca. 20 days at 20 °C). Therefore, larvae in the breeding experiment likely did not prolong development due to unfavorable conditions. Larvae also did not increase or decrease number of their instars, and they had to undergo three larval instars before maturation, which was also reported by *Kilian & Mądra (2015)*. Aggression or hostility was not observed between specimens, nor was there cannibalism as has previously been reported for this species (*Kilian & Mądra, 2015*). However, it was possible that it was missed due to the large number of larvae and adults in a box, close to one hundred. The photographing process was not so intrusive to be responsible for high mortality rates, and thus it is more likely that separation from other larvae and adults was probably the reason for the increased mortality. *Peck (1975)* mentioned that the cave adapted beetle *Ptomaphagus hirtus* (Tellkampf, 1844) (Leiodidae: Cholevinae: Ptomaphagini) needed soil from its cave of origin to successfully complete development. Soil bacteria probably played some part in this process, because specimens did not develop on autoclaved soil. In the experiments, it is possible that adults feeding along with larvae could have provided such bacteria. Another explanation could be that feeding of multiple individuals is much more effective or improves the quality of the food source. As a support for this hypothesis, I observed a very rapid growth of some fungi in Petri dishes with a single larva, but almost none in the observational study containers where a large number of individuals were feeding.

The methodology of egg extraction was changed in the second year because eggs were easily overlooked in the substrate, and beetles refused to lay their eggs in offered damp cotton wool balls or small pieces of paper. To prevent bias in recorded time, a dish rotation methodology was introduced, and adults stayed in the same dish only one day and then were moved to another. Those used dishes were then regularly searched for emerging larvae. The main issue with this approach (dish rotation) is that egg mortality could not be determined.

The mean development time decreased with increasing temperature (Fig. 6), except for L2 and L3 instars in the 25 °C treatment. Therefore, the optimal temperature of *S. watsoni* L2 and L3 instars may be between 21 °C and 25 °C (e.g., temperature with highest developmental rate where the specimen is still able to reach maturity). Optimal temperatures for lower stages may be higher, which agrees with the findings of *Engler (1981)*, who reported *S. watsoni* as a warm season species in contrast to some species of *Choleva* and *Catops* that prefer to breed during the winter season, and their optimal temperatures for development were below 16 °C (e.g., *Catops nigricans*, *Choleva agilis* and *Ch. elongata*). This hypothesis also agrees with the findings of *Ridgeway et al. (2014)* who reported that the optimal temperature for Afrotropic *Thanatophilus mutilatus* is between 14 and 25 °C.

Measuring development time for pupae was even more challenging due to the fact that they did not pupate close to the wall of the Petri dish. Searching for them was sometimes unsuccessful, and some specimens surprised us when they reappeared as adults, because they had been recorded as missing and presumed dead.

The methodology of measuring the size of the instars was based on continual observation of separated individuals, from egg until pupation, so stage-specific information was available regardless of their size. This approach differs from other studies with similar goals (*Velásquez & Viloria, 2010*; *Fratczak & Matuszewski, 2014*), where authors tried to estimate the stage of development based on the size of selected characteristics without prior knowledge of the true stage of the specimen. The latter approach can be problematic, because measured characteristics are correlated, and therefore larger larvae could be misidentified as a later instar. This bias would probably not affect the obtained mean values, but it would give a distorted picture of variation and ultimately give false confidence in determinations. This approach could be corrected if information about the risk of erroneous determination would be calculated beforehand (see *Merville et al., 2014*).

The data demonstrated that instars have some overlap in the head widths, especially true for the first and second instar. In these regions the uncertainty of determination based solely on the head width is very high, so the use of some additional morphological characteristics is in place. A first instar larva has only primary setae on its body and the head is without any colored spots, but after molting to the second instar, a secondary set of setae will emerge and a brown spot will appear on the head (light brown and not fully defined) (*Kilian & Mądra, 2015*). Setae are also present without any change on the third instar larvae, but the brown spot is much darker with sharp and well defined edge (*Kilian & Mądra, 2015*). Thus, chaetotaxy and pigmentation of the head can be used for the discrimination of all instars. The data provide developmental parameters for *S. watsoni* together with a new and reliable characteristic for instar determination. This species is so far the smallest necrophagous beetle with a known thermal summation model. The developmental characteristics provided in this study will help to more accurately estimate the PMImin.

## ACKNOWLEDGEMENTS

I would like to thank A Honěk and P Saska for sharing their insight about beetle development and construction of thermal summation models. I am also grateful to Jan Růžička and Max Barclay who provided many valuable comment and language corrections. This research would not have been possible without the help of my students from the Czech University of Life Sciences Prague: T Račáková, J Pšajdl and M Slachová, who took care of the experiment at times I could not.

### Funding

The project was supported by the Internal Grant Agency of Faculty of Environmental Sciences, CULS Prague (no. 20154221) and by the Ministry of the Interior of the Czech

Republic (no. VI20152018027). The funders had no role in study design, data collection and analysis, decision to publish, or preparation of the manuscript.

**Grant Disclosures**

The following grant information was disclosed by the author:
Internal Grant Agency of Faculty of Environmental Sciences, CULS Prague: 20154221.
Ministry of the Interior of the Czech Republic: VI20152018027.

**Competing Interests**

The authors declare there are no competing interests.

**Author Contributions**

- Pavel Jakubec conceived and designed the experiments, performed the experiments, analyzed the data, contributed reagents/materials/analysis tools, wrote the paper, prepared figures and/or tables.

**Data Availability**

Figshare: https://figshare.com/articles/Larval_development_of_Sciodrepoides_watsoni_Coleoptera_Leiodidae_Cholevinae_/1531668

Github: https://github.com/jakubecp/sciodrepoides.

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
