# Peer review of "Thermal summation model and instar determination of all developmental stages of necrophagous beetle, Sciodrepoides watsoni (Spence) (Coleoptera: Leiodidae: Cholevinae)"

_PeerJ, doi:10.7717/peerj.1944_

## Round 0.1 · original submission · Major Revisions

Please address the comments of the reviewers, particularly reviewer #1 who has a number of concerns. Please note that using the head capsule to determine larval age is quite a common practice in Entomology.

Reviewer 1 ·

Basic reporting

The writing in English was remarkably clear (although in some instances did not conform to usual practice).
The background for how the work fits into a broader field of knowledge is not sufficient. The author "... proposes head width as a new character for larval instar determination ..." (line 18-20), but this is not a new character, unless the author is proposing it only for S. watsoni. Dyar's law was proposed in 1890 (H. G. Dyar, 1890. The number of molts of lepidopterous larvae. Psyche 4: 420-422), suggesting that head width increases by a ratio which is constant for a species. Recognition of instars by head width is commonly done by entomology graduate students, and published literature has revealed that Dyar's law is often useful, although exceptions to the constancy of the ratio are not uncommon. The author's argument that head width has advantages as a character is intelligent and valid, but not novel.
An important finding is that chaetotaxy of the 1st instar is rudimentary compared to other instars. Is this a new finding? This should be presented clearly in the context of the field of knowledge of chaetotaxy of larvae of this species and group (the family or Coleoptera in general).
The relatively low optimal temperature for development of this species is surprising to me, although it may be consistent with temperature in its geographic range. I would appreciate discussion of optima for species having similar ranges to S. watsoni.
I am uncertain that the paper is a sufficient "unit of publication," because of the extremely high mortality (>80%) for 3rd instar larvae and for the pupae. Due to the challenges of the study, it may be rightfully judged that presentation of adequate data for a "validation-ready" model for forensics should be a separate study, but I suggest that at least some preliminary experimental data should be presented in two important areas, the possible cause of the mortality and a comparison of the development of isolated larvae with development under natural conditions.
Insight into a possible cause of the mortality is provided in lines 222-229, suggesting that adults could provide bacteria that increase "... the quality of the food source." This is both fascinating and disturbing, since the number of larval instars is increased by prolonged development due to poor nutrition in some coleopterans. At a minimum, I would think that the typical range of head capsule widths for mature larvae in the field (the cast larval head capsules are likely to be present near the pupae) should be reported. Another valuable addition could be preliminary investigation of pre-treatment of the food with short term feeding by adult males (used to avoid oviposition).

Experimental design

The research question is inadequately or incorrectly defined. It think that the research is the investigation of methods that may bring us closer to development of a model of development of this beetle for forensics. The present paper clearly does not present a model which is ready for forensic application, and this has to be made clear in the abstract and elsewhere.
The investigation was conducted to a high technical standard and described with sufficient information to be reproducible, and is reported with high ethical standards. However, I suggest that the typical handling of larvae for photo documentation be described more fully.
Since the usefulness of photo documentation for estimation of head width was a major finding of the paper, it would have been interesting to have a comparison showing how similar these estimates are to those provided by a micrometer scale in the eyepiece of a microscope to measure head width of larvae that have been immobilized by chilling or other methods.
None of the experimental temperatures were low enough to prevent development, and therefore, the developmental threshold has not been established.

Validity of the findings

The conclusions about data are valid. In my opinion, however, it must be made clear that the model is unlikely to provide useful predictions of development in the field. I base my opinion on high mortality data and on my belief that it is very likely that development was abnormally slow. The methodology must be improved, if the model is going to have a reasonable chance of validation. Any predictive model requires validation before its predictions can be used with confidence, and in this case, I assume that validation would be comparison of predictions with data on presence of developmental stages at intervals after infestation of food under natural conditions.

·

Basic reporting

Please, add more sufficient background about previous studies in which head width was used to determine larval instars. Therer are many studies where the haed width was measured and difference of head width among larval instars are commonly used.
So, it is not a new character for larval determination,I suggest to underline a practical aspect of this character in forensic entomology as well as the importance of other morphological features in determining of larval instars in legal investigation.

Experimental design

See comments on the attached annotated manuscript

Validity of the findings

See comments on the attached annotated manuscript

Additional comments

See comments on the attached annotated manuscript

---

## Round 0.2 · Major Revisions

While you have made a great effort to address the reviewer's concerns, there are still some problems, the most pressing is in editing for English. I have provided suggestions on editing in the PDF attached-the journal will provide you with a word version that will be easier to work with.

The word "our" and "we" is used in the abstract and article, but yet you are the only author. I have provided editing that removes these references. Also, the English word “character” is used throughout, but it seems more appropriate to use the word “characteristic”. In addition, some of the language is more “casual” rather than scientific, and I have made some suggestions. I have also removed several sentences that were unclear or that were over-reaching in explaining the implications of the data.

As far as the reviewer's comments, both asked you to provide reference to the common practices of instar determination by head capsule width, which still is lacking in the manuscript. I understand your rebuttal that it is not common in forensic entomology and is not available for this species. However, the reviewers asked that you reference the common practice of head capsule measurement, and even provided a reference (ie., reference to Dyar’s law).

The data is interesting and presented in nice graphical formats. I hope that you will find the editing useful so that the readership can appreciate the full impact of your research.

---

## Round 0.3 · Minor Revisions

I am sorry but one of the reviewers still asks for some clarifications. I am not so concerned with most of the comments, as I have edited the manuscript (attached) for typos and English, but the reviewer would like specific info on how the measurements were made. Perhaps this can be clarified in the very nice photos in Fig. 3, where you indicate that the arrow (a) denotes the head capsule measurements that you made (putting this in the figure legend, and maybe also in the text on line 177).

Thank you for your patience and diligence in making these changes. We are almost ready for acceptance, with these minor corrections.

·

Basic reporting

Structure conforms to PeerJ standard,

Unambiguous, professional English language in spie of few sentences (in separate file)
Introduction show context (but still well seen references to other studies, where head width was used)

Literature well referenced & relevant (lack literature about using head width)

Figures and tables are relevant, high quality, well labelled & described.

Experimental design

Generally, methods described with sufficient detail & information to replicate. BUT
How author decided in case of extreme of measurements, which instar it is? It should be given in methods. I asked about it previously.
Author did not provide detail information in which points, exactly, head width have been measured. I asked about it previously.

Research question well defined, relevant.

Validity of the findings

The study should help using S. watsoni immatue stages in legal investigations in forensic entomology.

Additional comments

Dear Author, you still does not provide detail information in which points, exactly, head width have been measured and still do not proved practical way how to determine instars in cases of extreme values of head width (for example, prepare slides, use precise morphological characters, or other measurements) or may be, in legal investigation, just omit such extreme data? I can’t find in result or discussion exact explanation, summary of this method and providing combination of size based and morphological characters. How author decide in case of extreme of measurements, which instar it is? It should be given in methods. I asked about it previously.

See the attached PDF for detailed edits.

---

## Round 0.4 · accepted · Accept

Thank you for your diligence and hard work to address the reviewer comments.